# Development and Biochemical Characterization of Quorum Quenching Enzyme from Deep-Sea *Bacillus velezensis* DH82

**DOI:** 10.3390/microorganisms13081717

**Published:** 2025-07-22

**Authors:** Xiaohui Sun, Jia Liu, Ying Yan, Suping Yang, Guangya Zhang, Hala F. Mohamed

**Affiliations:** 1College of Chemical Engineering, Huaqiao University, Xiamen 361021, China; jialiu0505@163.com (J.L.); yanying@stu.hqu.edu.cn (Y.Y.); yangsuping@hqu.edu.cn (S.Y.); zhgyghh@hqu.edu.cn (G.Z.); 2Department of Botany and Microbiology, Faculty of Science, Al-Azhar University (Girls Branch), Cairo 11651, Egypt

**Keywords:** quorum quenching, *N*-acyl homoserine lactonase, enzyme activity, deep sea, *Bacillus velezensis*

## Abstract

Quorum quenching (QQ) is of interest for potential application as a sustainable strategy for bacterial disease control via communication interruption. The QQ enzyme can be used as a good alternative antagonist to combat antibiotic abuse and bacterial resistance. Here, genomic DNA sequencing was performed on *N*-acyl homoserine lactonase from the deep-sea strain *Bacillus velezensis* DH82 with Cluster of Orthologous Groups of proteins (COGs) annotation. The homologous sequences with β-lactamase domain-containing protein were predicted to be potential QQ enzymes and were cloned and expressed to study their quorum quenching properties by comparing them with the reported enzyme AiiA_3DHB_. The experimental results of enzyme activity analysis and steady-state kinetics, as well as enzyme structure and substrate docking simulations and predictions, all consistently demonstrated that YtnP_DH82_ presented superior enzyme structural stability and higher degradation efficiency of *N*-acyl homoserine lactones than AiiA_DH82_ under the effects of pH, and temperature, and performed better on short -chain and 3-O-substituted AHSLs. The findings revealed the structural and biochemical characterization of YtnP_DH82_ from the deep sea, which provide the capacity for further application in sustainable aquaculture as an alternative to antibiotics.

## 1. Introduction

Intensive and high-stocking-density aquaculture generates a high output of aquatic animals but also leads to a high risk of pathogen infection and frequent bacterial disease in aquatic animals. The traditional technique for solving this problem involves using antibiotics or chemical compounds to inhibit pathogens and prevent bacterial infection. However, the abuse of antibiotics and drug residues can cause bacterial drug resistance, which has increasingly raised concerns about food safety and environmental protection; thus, an alternative strategy for biological disease control is required to support the development of sustainable aquaculture.

Aquatic pathogens are mainly Gram-negative bacteria, such as *Vibrio* sp., *Aeromonas* sp., and *Pseudomonas* sp., that communicate with a common auto-inducer, *N*-acyl-L-homoserine lactone (AHL), for quorum sensing (QS) regulation [1]. The QS inhibitor [2,3], for example, the quorum quenching (QQ) enzymes, which degrade the AHL signaling molecules, including AHL-lactonase, -acylase, and -oxidoreductases, has been applied as an alternative antibacterial reagent with the advantages of precise targeting control and mitigation of bacterial antibiotic resistance. However, the application of QQ enzymes is still limited by their low catalytic efficiency and enzyme stability in comprehensive aquacultural environments, which are normally affected by high salt, multiple iron, pH, and temperature ranges.

Compared to land resources, marine microorganisms, especially those from the deep surface, normally contain homologous genes with low similarity to currently known sequences and exhibit specific functions [4,5]. In previous work, *Bacillus velezensis* DH82 was isolated from underlying sea water of the Western Pacific Yap trench at a depth of 6000 m, and was identified, using the quorum quenching function, as affecting the biofilm-forming ability and pathogenicity of *P. aeruginosain* [6] and *V. parahemolyticus* [7]. For further utilization of this strain, in this study, the potential QQ enzymes were predicted by performing whole-genome sequencing and COG annotation according to the homological sequence of the serine active site of *N*-acyl homoserine lactonase AiiA (EC:3.1.1.81) [8,9,10]. The predicted QQ enzymes were engineered for heterologous expression in *E. coli* to study their enzyme activity on AHL degradation and their stability against environmental factors.

## 2. Materials and Methods

### 2.1. Bacteria, Plasmids, and Reagents

The *B. velezensis* DH82 strain (GenBank: MK203035) was isolated from sea water samples taken from the Western Pacific Yap trench at a depth of 6000 m, and was kindly provided by the Third Institute of Oceanography (Xiamen, China). The competent cells of the *E. coli* DH5α and *E. coli* BL21 strains, Isopropyl-β-D-thiogalactopyranoside (IPTG), and Kanamycin, were purchased from Transgen (Beijing, China). The AHL reporter operon of LuxR-*P_luxI-lacO_*-RFP was provided by Xiamen University (Xiamen, China) [11]. *N*-Acyl-L-homoserine lactone hydrolase (PDB: 3DHB) from *B. thuringiensis* (GenBank: AY943832), named AiiA_3DHB_, was used as the positive control to analyze the activity of the predicted QQ enzyme.

The plasmid pET28a expression vector was purchased from Novagen (Houston, TX, USA) (Cat. 69864-3). The plasmid miniprep kit (Cat. GMK5999) and gel extraction kit (Cat. D2500-02) were purchased from Promega (Beijing, China). The AHLs, C_6_-HSL (Cat. 56395), 3-oxo-C_6_-HSL (Cat. K3255), C_4_-HSL (Cat. 09945), 3-OH-C_4_-HSL (Cat. 74359), 3-oxo-C_10_-HSL (Cat. O9014), and 3-oxo-C_12_-HSL (Cat. O9139), were purchased from Sigma-Aldrich (Shanghai, China).

### 2.2. Genomic DNA Sequencing

The bacterial culture of DH82 was incubated at 37 °C with shaking at 180 rpm for 12 h and was centrifuged at 8000 rpm for 10 min to harvest the bacterial pellets. Then, 100 ng of the genomic DNA of the strain DH82 was extracted from the pellets by using the CTAB method, and randomly digested into fragments of less than 500 bp by ultrasonication using the Covaris S220 to establish the cDNA library. The quality of cDNA was assessed using an Agilent 2100 Bioanalyzer (Agilent Technologies, Palo Alto, CA, USA), while the quantity of cDNA was assessed using a Qubit 3.0 fluorometer (Thermo Fisher Scientific, Waltham, MA, USA). Sequencing was performed using the Illumina Hiseq×10 and PacBio RSII by Majorbio (Shanghai, China).

### 2.3. Gene Cloning

The sequences of AiiA_DH82_ and YtnP_DH82_ were amplified from the genomic DNA of DH82 using PCR and the primers listed in Table 1, which were synthesized by Sangon Biotech (Shanghai) Co., Ltd. (Shanghai, China). The PCR products were digested with the restriction enzymes *Nde*I, *Xho*I, and *EcoR*I (Takara, Beijing, China) and subsequently ligated to the multiple cloning sites of the pET28a vector using T4 ligase (Takara, Beijing, China). The engineered expression clone, which was driven by the T7 promoter and framed with 6× Histidine (His) at both N- and C-terminal ends to ensure compatibility with the target enzyme, was transferred to *E. coli* BL21 competent cells for protein expression.

The AHL reporter operon of LuxR-*P_luxI-lacO_*-RFP [11] was digested using restriction enzymes *Nde*I and *Hpa*I, and then ligated into the pET28a plasmid. The re-engineered reporter plasmid was then transferred to *E. coli* BL21 competent cells to construct the reporter strain (named LuxR-RFP for short) in this study for rapid detection of AHLs.

### 2.4. Bacterial Culture and Protein Expression

The bacteria were cultured using Luria–Bertani (LB) media (10 g/L tryptone, 5 g/L yeast extract, and 10 g/L NaCl, pH 7.0) containing 50 µg/mL kanamycin with shaking at 180 rpm at 37 °C. Then, 0.2 mM IPTG was inoculated after 2 h of incubation with the bacterial culture to induce the protein expression. The overnight cultured bacteria were washed and concentrated at 5:1 with PBS (pH 7.0), ultrasonically broken (operate for 3 s at 300 W then break for 6 s, and repeat 60 times) on ice, centrifuged at 4 °C at 10,000 rpm for 15 min, and the supernatant was harvested. The extracted crude enzyme was resuspended using lysis buffer (300 mM NaCl and 50 mM NaH_2_PO_4_ (pH 7.4)), then washed with imidazole elution buffer (300 mM NaCl, 200 mM imidazole, and 50 mM NaH_2_PO_4_ (pH 7.4)). High-affinity NI-NTA chromatography was used to purify the His-tagged enzyme. The purified enzyme was filter-sterilized and analyzed using SDS-PAGE. The concentration of protein was quantified using the Bradford assay.

### 2.5. In Vitro Rapid Assessment of AHLs Level

Next, 100 µL of filter-sterilized AHL solution (working concentration, 400 mM) was mixed with 100 µL of purified enzyme in triplicate, using PBS as a negative control at the same volume. The enzyme-AHL mixtures were incubated at 28 °C for 45 min, followed by an immediate addition of 10% SDS to stop the reaction. The reporter strain was incubated overnight in LB media (containing 50 µg/mL Kanamycin) at 37 °C with shaking at 200 rpm, and then inoculated with the above enzyme-AHL mixture, incubated at 25 °C with shaking at 180 rpm for 12 h, centrifuged at 8000× *g* at 4 °C, and resuspended in an equal volume of PBS. A fluorescence intensity of 620 nm (excitation wavelength of 584 nm) and an optical density of 600 nm (*OD*600) were measured using a Tecan Infinite M200 Pro. microplate reader (Tecan Group Ltd., Grödig, Austria) to determine the residual AHL levels in the samples.

### 2.6. Quantitative Analysis of Enzyme Activity on AHL Degradation

We analyzed 200 µL of filter-sterilized 3-oxo-C_6_-HSL at final concentrations of 0.05, 0.5, 1.0, 1.5, and 2.0 µM using a C18 chromatographic column (Agilent, 1.8 μm, 2.1 × 50 mm) on ultra-performance liquid chromatography-quadrupole time of flight mass spectrometry (UPLC-QTOF MS, Agilent, 1290-6545) to generate a standard curve, with AHL concentrations determined by the peak area at the mass-to-charge ratio of 214.

#### 2.6.1. Characterization of Enzyme Activity

Filter-sterilized 3-oxo-C_6_-HSL (final concentration of 300 µM) was added with 100 µL of 0.5 mg/mL purified lactonase in 1 mL reaction volumes and incubated at pH 7.5 at 18, 25, 37, 45, 55, and 65 °C for 45 min. Sterilized water and non-treated AHL served as negative and positive controls, respectively. Reactions were terminated by heating at 100 °C for 10 min, and the residual AHLs were quantified using a C18 column on UPLC-QTOF MS, with the standard curve used to analyze the temperature effects on the enzyme activity.

Similarly, enzyme-AHL mixtures were incubated at 28 °C under pH 2.0, 3.0, 4.0, 5.0, and 6.0 for 45 min, with residual AHLs quantified to evaluate the pH effects.

#### 2.6.2. Kinetic Analysis of Enzyme Activity

We mixed 100 µL of filter-sterilized AHLs at 200, 250, 300, and 350 µM with 100 µL of 0.5 mg/mL purified enzyme before incubating them at pH 6.5 and 28 °C for 45 min. Reactions were terminated at 100 °C for 10 min, and residual 3-oxo-C_6_-HSL was analyzed using a C18 column on UPLC-QTOF MS. Residual AHLs were determined by the peak area at *m*/*z* 214. Lineweaver–Burk plots were established to calculate the Km value and reaction rate (V_max_).

### 2.7. Three-Dimensional Structure Simulation and Functional Prediction

Three-dimensional (3D) enzyme structures were simulated and predicted using the I-TASSER server [12]; the simulated models with a high confidence score were analyzed using VMD 1.8.3 software to identify ligand-binding regions and active sites involved in AHL degradation for functional prediction.

### 2.8. Analysis of Enzyme-Substrate Docking

Molecular docking was simulated using the CHARMm-based molecular dynamics (MD) scheme in Discovery Studio 2019 [13]. Random ligand conformations were generated using high-temperature MD, translated into the enzyme binding site with rigid-body rotations, followed by simulated annealing. Docked ligand poses were recorded with the CDOCKER score, including internal ligand strain energy and receptor–ligand interaction energy (higher values indicate more favorable binding).

### 2.9. Statistical Analysis

T-tests were applied to compare enzyme activities, which were analyzed using GraphPad Prism 6. The *p*-value indicated differences between groups.

## 3. Results

### 3.1. Prediction and Phylogenetic Tree Analysis of Quorum Quenching Enzyme

According to the COG annotation from the sequencing results, as shown in Figure 1, a panel of proteins containing a conserved β-lactamase domain—which were consistent with the reported QQ lactonase families—were predicted to be QQ-related proteins due to their homologous serine active sites with potential AHL-binding ability. Metal-binding sites were categorized as metal-dependent hydrolases. Protein 4079 showed 91.6% consensus with the quorum-quenching lactonase AiiA from *B. thuringiensis* (PDB ID: 3DHB) [9,10,11], and was 98.40% homologous to AiiA from *B. cereus* Y2 (Swiss-Prot: Q08GP4.1) [14].

Protein 0540 was 79% homologous to YtnP from *B. subtilis* 168 (Swiss-Prot: O34760.2) [15], 26.14% to Y2-AiiA from *B. cereus* Y2 (Swiss-Prot: Q08GP4.1) [14], and 27.84% to AiiB from *Agrobacterium fabrum* C58 (Swiss-Prot: A9CKY2.1) [16].

Other predicted proteins were not reported as QQ enzymes: protein 0114 was 99.53% homologous to YqgX from *B. velezensis* G341 [17]; protein 1459 was identified as putative *Bacillaene* biosynthesis zinc-dependent hydrolase and was a 100% match to BaeB from *B. velezensis* UCMB5033 (GenBank: AIU81842.1; UniPro: S6FKG4) [18]; protein 1509 was a 59.1% match to probable metallo-hydrolase YqjP from *B. subtilis* 168 (UniPro: P54553) [15]; protein 1800 was 28.64% homologous to MBL fold metallo-hydrolase YflN [19] from *Paenibacillus flagellates* (UniProt: A0A2V5K1S6).

Phylogenetic analysis using an NCBI-BLAST (ElasticBLAST 1.4.0) with maximum likelihood clustering is shown in Figure 2. The results revealed that proteins 0114, 1459, and 1800 were 100% homologous to metallo-β-lactamases, while protein 1509 showed more than 70% similarity to *Bacillus*-derived sequences. Proteins 4079 and 0540 shared 93% and 88% similarity with AiiA from *B. cereus* Y2 and YtnP from *B. mojavensis*, respectively, and were named AiiA_DH82_ and YtnP_DH82_ for further biological characterization.

### 3.2. Construction of Engineered Enzyme and Protein Expression

The expression clones of His-tagged AiiA_DH82_ and YtnP_DH82_ were constructed on the pET28a plasmid and expressed in *E. coli* BL21. The sequence of the expression clones was confirmed via sequencing by Sangon Biotech (Shanghai) Co., Ltd.

The heterologously expressed proteins were analyzed using SDS-PAGE as shown in Figure 3. The His-tagged AiiA_DH82_ at 31.99 kDa and YtnP_DH82_ at 35.82 kDa were observed in Lanes 3 and 4, respectively, whereas the crude enzyme extraction of AiiA_DH82_ was observed in Lane 1 and YtnP_DH82_ in Lane 2. The concentrations of harvested purified AiiA_DH82_ and YtnP_DH82_ were 0.594 mg/mL and 0.513 mg/mL, respectively, according to a Bradford assay.

### 3.3. Identification of Quorum Quenching Capacity on AHL Degradation

The activity of recombinant enzymes was assessed according to AHL degradation against a series of AHLs. Fluorescence intensity generated from LuxR-RFP was used to determine the residual AHLs in the reaction. As shown in Figure 4, both AiiA_DH82_ and YtnP_DH82_ were observed with function on 3-oxo-C_6_-HSL, C_6_-HSL, 3-OH-C_4_-HSL, C_4_-HSL, 3-oxo-C_12_-HSL and 3-oxo-C_10_-HSL.

### 3.4. 3D Structural Simulation and Analysis

#### 3.4.1. 3D Structure of Lactonase

The 3D structures of AiiA_DH82_ and YtnP_DH82_ were simulated using I-TASSER, which employed structural alignment to match the structures in the PDB library.

YtnP_DH82_ was predicted with an average distance of all residue pairs in two structures (RMSD) at 4.1 ± 2.7 Å (highest C-score at 0.97 and TM-score at 0.85 ± 0.08). The model of the most typical structure (Figure 5A) is presented with a substrate-binding pocket formed by the α-helix (labeled in purple), the Zn^2+^-binding site (H111, H113, H191, and D212), the FEO (µ-oxo-diiron) binding site (H111, H113, D115, H116, H191, D212, and H257), and the active site at residue D115. As shown in the magnified view presented in Figure 5B, the first Zn^2+^-binding site is composed of nitrogen atoms on H111 and H191, and an oxygen atom on D212; the second Zn^2+^-binding site is composed of a nitrogen atom on H116 and H257, and an oxygen atom on D115. Both Zn^2+^ ions bind an oxygen atom after the opening of the ester ring on AHL.

The structure of AiiA_DH82_ is depicted in Figure 5C and magnified in Figure 5D, with an average distance of all residue pairs in two structures (RMSD) at 2.6 ± 1.9 Å (highest TM-score at 0.998), containing a flexible substrate pocket, a Zn^2+^-binding site (residues H104, H106, H169, and D191), a substrate pocket (C14, F64, T67, F68, I73, H106, F107, D108, H109, E136, H169, D191, Y194, and H235), and an active site at residue D108.

The structural simulation indicates that YtnP_DH82_ is more stable with a fixed substrate pocket with an α-helix, and that AiiA_DH82_ exhibits higher activity with a more flexible substrate pocket, which was verified in further experiments under the challenges of different temperatures and pH.

#### 3.4.2. Capacity of Lactonase-AHL Docking

The capacity of enzyme-substrate docking was analyzed for the top 10 docked structures of the enzyme with a ligand pose (Table 2). A higher CDOCKER score means a greater tendency and a higher possibility for substrate binding. The results demonstrated that YtnP_DH82_ showed higher affinity for short-chain AHLs (notably C_6_-HSLs), while AiiA_DH82_ preferred long-chain substrates such as C_10_-HSL and C_12_-HSL. YtnP_DH82_ showed enhanced activity toward 3-O-substituted C_4_- and C_6_-HSLs, while such a tendency was not observed in AiiA_DH82_.

#### 3.4.3. Prediction of Ester-Hydrolysis Activity on AHLs

The molecular dynamics were further analyzed using ligand poses and the average CDOCKER score (Figure 6) to predict their ester-hydrolysis activity against AHLs, according to the distance between the AHLs’ ester group and the active sites. The ligand poses showed that YtnP_DH82_ was closer to 3-oxo-C_6_-HSL (5.85 Å) than AiiA (11.09 Å), which indicated that YtnP_DH82_ had higher hydrolysis activity. However, AiiA_DH82_ was closer to C_6_-HSL and 3-oxo-C_12_-HSL than YtnP_DH82_, suggesting that AiiA_DH82_ had superior activity toward these substrates.

### 3.5. Qualification of AHL Degradation Activity

The classical AHL, 3-oxo-C_6_-HSL, was further used to quantitatively assess the enzyme activity of lactonases. A lower residual amount of AHLs indicates that the lactonase has a stronger degradation activity.

As shown in Figure 7A, after a 45-min catalytic reaction, AiiA_DH82_ exhibited greater adaptability (with degradation activity all above 89.95%) across a broad pH range (2.0–6.0). Its activity peaked at pH 6.0 (95.61% degradation) with 1318.37 nM of residual AHL. In contrast, YtnP_DH82_ was sensitive to strong acidic conditions but demonstrated enhanced stability as the pH increased to 6.0, at which point its AHL-degradation activity reached a maximum of 99.99% with only 4.378 nM of residual AHL remaining.

Figure 7B illustrates that both YtnP_DH82_ and AiiA_DH82_ reduced the AHL concentration to below 5 nM. Furthermore, YtnP_DH82_ showed higher stability and activity within the temperature range of 18–65 °C, with degradation rates against 3-oxo-C_6_-HSL all exceeding 99.99% and a peak value of 99.9987% at 18 °C. In comparison, AiiA_DH82_ showed lower activity and a declining trend, with its highest activity of 99.9955% degradation achieved at 18 °C.

### 3.6. Steady-State Kinetics Characterization of Enzyme

The kinetic characterization of AiiA_DH82_ and YtnP_DH82_ was further performed using the substrate of 3-oxo-C_6_-HSL, as shown in Table 3, according to the Lineweaver–Burk equation obtained from the standard curve. The Km value of YtnP_DH82_ was 15.33 mmol/L, and the V_max_ was 0.33 mmol/min, while the Km of AiiA_DH82_ was 7.8 mmol/L and 0.18 mmol/min.

## 4. Discussion

Bacteria produce various molecular signals to communicate through the QS system [20]. However, it remains unknown whether bacteria have evolved to produce signals via QS for communication or to degrade signals via QQ for competition [21,22]. Researchers have attempted to reveal the novel QS signals and the regulatory mechanisms of QS systems [1], as well as to discover novel QQ enzymes [23] to interfere with bacterial QS by degrading the relevant QS signals [24].

AiiA was initially classified as belonging to the metallo-β-lactamase superfamily due to its conserved HXHXDH region. Another conserved sequence, GHXXGX, was later identified in AiiA (PDB: 3DHB) from *B. thuringiensis* [10]. Both sequences constitute Zn^2+^-binding sites: the HXHXDH region binds Zn^2+^ and Tyr194 stabilizes the charge of the tetrahedral AHL substrate, thereby promoting nucleophilic attack; in contrast, the GHXXGX sequence binds Zn^2+^, stabilizing the charge of the alkoxide formed after the AHL ring opens to break the C-O ester bond [25,26].

YtnP_DH82_ was also classified as belonging to the metallo-β-lactamase superfamily, containing both the conserved HXHXDH and GHXXGX regions. The enzyme structure reveals a conserved active site with Asp115 substitution, similar to AiiA-like lactonase but with a distinctive feature—a “kinked” α-helix that forms part of a closed binding pocket that provides affinity and enforces selectivity for AHL substrates, similar to previously characterized AHL lactonase AidC [27].

The structural features of both lactonases were reflected in the enzyme activities on AHL degradation. For 3-oxo-C_6_-HSL, a signal molecule reported to be produced by numerous Gram-negative bacteria, and especially the primary signal for *Vibrio* sp. Pathogens [28], AiiA_DH82_ has a Km value of 7.8 mM, showing no significant difference from other reported wild-type AiiAs [10]. In contrast, YtnP_DH82_, despite sharing the same active site, has a more extensive α-helix structure, resulting in lower substrate affinity with a Km of 15.33 mM. Additionally, the α-helical feature demonstrated its advantage for enzyme stability, as evidenced by a broader tolerance to changes in temperature. Additionally, YtnP_DH82_ exhibits higher kcat and kcat/Km values than AiiA_DH82_, which is consistent with the ligand pose of the target C-O bond within the substrate pocket relative to the active site of Asp substitution.

For AHL compounds, differences in acyl chain length (4 to 20 carbons) or substituents at the 3-position of the *N*-acyl side chain (no substituent, keto, or hydroxy) all influence enzyme activity in substrate hydrolysis. The activities of both lactonases correlate positively with the substrate’s acyl chain length. Furthermore, YtnP_DH82_ showed higher activity toward short-chain substrates such as C_4_ and C_6_, whereas AiiA_DH82_ performs better with longer-chain substrates such as C_10_-HSL and C_12_-HSL. Substrate effects on enzyme activity also differ slightly, as YtnP_DH82_ has higher activity toward substrates with oxygen-containing substituents (for example, it performs better on 3-oxo-C_6_-HSL than on C_6_-HSL), while AiiA_DH82_ showed no significant activity toward short-chain substrates.

The QQ efficiency of lactonase was also evaluated using a quantitative analysis of the residual AHL concentrations. Considering the alkaline hydrolysis character of AHLs at pH values above 7 [29], the activity of lactonases was only accessed in the range of acidic conditions used in this study. Bacterial QS systems are threshold-dependent, relying on signal accumulation and removal [30]; for instance, the receptor protein LuxR requires a threshold concentration of 5 nM for the canonical signal molecule 3-oxo-C_6_-HSL [30,31]. Therefore, YtnP_DH82_ demonstrates an advantage over AiiA_DH82_ in terms of QS regulation via 3-oxo-C_6_-HSL degradation, as evidenced by its higher degradation rates across different pH and temperature conditions, which indicates the greater potential and broader applicability of YtnP_DH82_ in regulating the AHL-mediated QS pathway in Gram-negative bacteria.

## 5. Conclusions

In summary, both AiiA_DH82_ and YtnP_DH82_ from the *B. velezensis* DH82 strain belong to the metallo-β-lactamase superfamily. They share the same conserved domains and active sites for AHL hydrolysis, but differ in the structural features of substrate pockets. Herein, AiiA_DH82_ demonstrated similar biochemical properties to other reported lactonases and showed greater adaptability to pH fluctuations and higher activity toward long-chain substrates due to its more flexible substrate pocket in the enzyme structure. In contrast, the α-helix in YtnP_DH82_ confers substrate affinity and enforces selectivity, enabling it to perform better in degrading short-chain AHLs such as 3-oxo-C_6_-HSL. Additionally, YtnP_DH82_ displayed higher catalytic activity and greater stability under varying environmental conditions, such as temperature. These findings clarify the structural features and biochemical characterization of QQ enzymes in the probiotic *B. velezensis* DH82 strain and provide an experimental basis for the further application of YtnP_DH82_ in regulating AHL-mediated QS pathways.

## Figures and Tables

**Figure 1 microorganisms-13-01717-f001:**
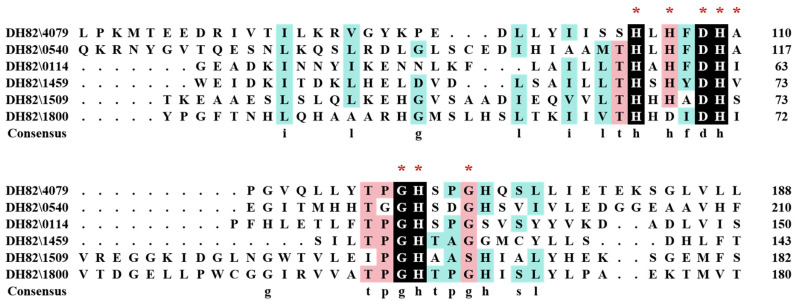
Multiple alignment of β-lactamase domain-containing protein superfamily from DH82 strain. The residues that were completely conserved are represented by black shading. The residues that were not completely homologous were labelled in pink and light blue. The conserved regions HXHXDHA and GHXXGX were labelled with a red *.

**Figure 2 microorganisms-13-01717-f002:**
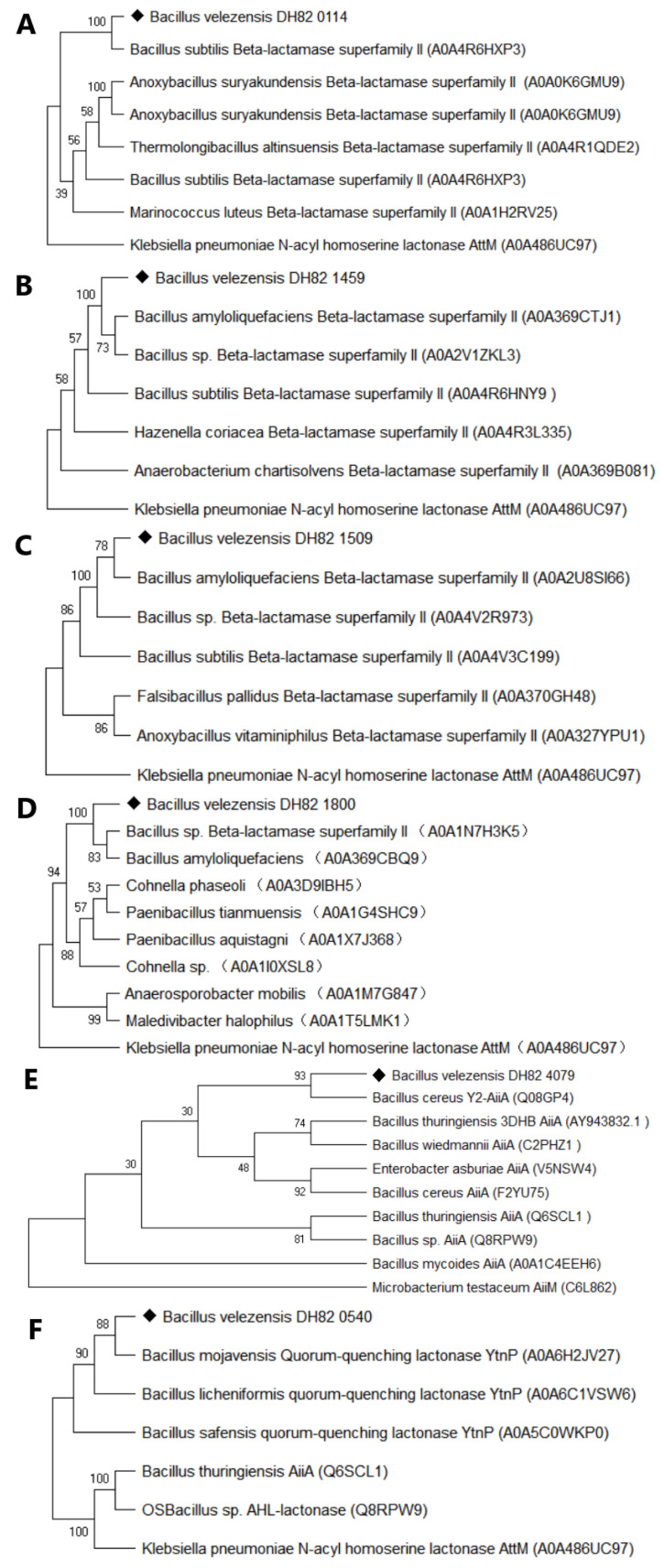
Phylogenetic analysis of predicted enzymes. The nucleotide sequence of each predicted QQ enzyme was analyzed using NCBI-BLAST. Proteins 0114, 1459, 1509, 1800, 4079, and 0540 were respectively present in panels (**A**–**F**) labelled with a ◆.

**Figure 3 microorganisms-13-01717-f003:**
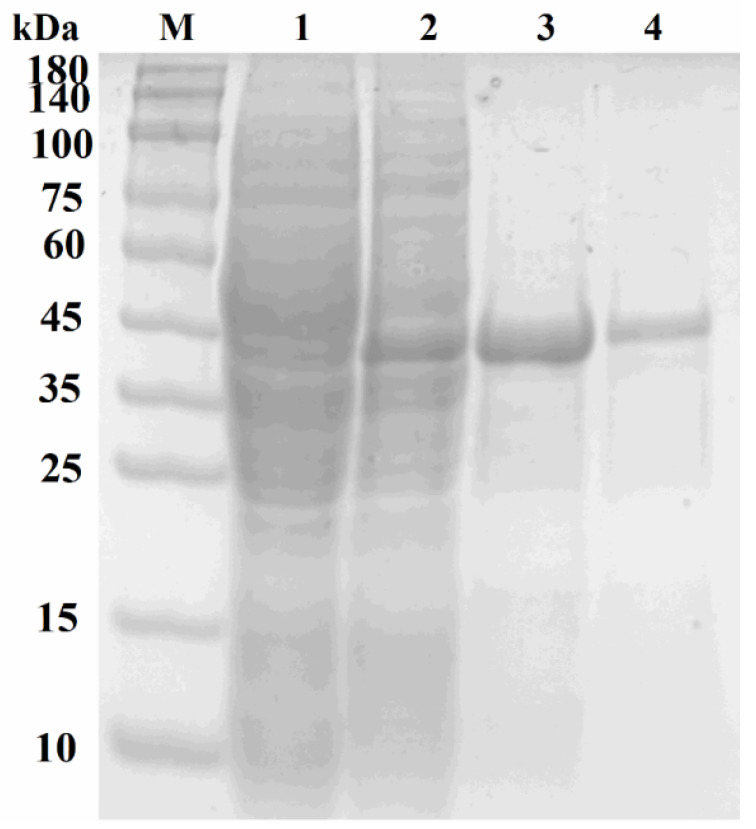
SDS-PAGE analysis of engineered AiiA_DH82_ and YtnP_DH82_. The engineered lactonases were expressed overnight at 37 °C in *E. coli* under induction with 0.4 mM IPTG; the expressed proteins were harvested and analyzed using SDS-PAGE. Lane M, marker; Lane 1, crude extraction from *E. coli::*pET28a/AiiA_DH82_; Lane 2, crude extraction from *E. coli::*pET28a/YtnP_DH82_; Lane 3, purified AiiA_DH82_; Lane 4, purified YtnP_DH82_.

**Figure 4 microorganisms-13-01717-f004:**
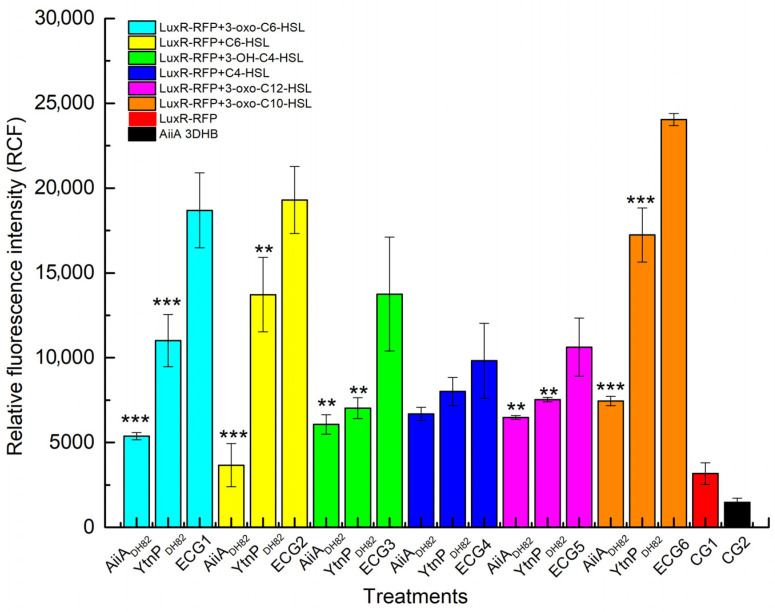
QQ activity on AHL degradation. Activity of the predicted proteins was determined by the relative fluorescence intensity of LuxR-RFP. Degradation of 3-oxo-C_6_-HSL (cyan), C_6_-HSL (yellow), 3-OH-C_4_-HSL (green), C_4_-HSL (blue), 3-oxo-C_12_-HSL (purple), and 3-oxo-C_10_-HSL (orange) by AiiA_DH82_ and YtnP_DH82_, compared with untreated AHLs as experimental control groups (ECGs). Controls include reporter without AHL (CG1 in red) and LB only (CG2 in black). Statistical analysis results are presented, with significant differences indicated by *** where *p* < 0.01 and ** where *p* < 0.05.

**Figure 5 microorganisms-13-01717-f005:**
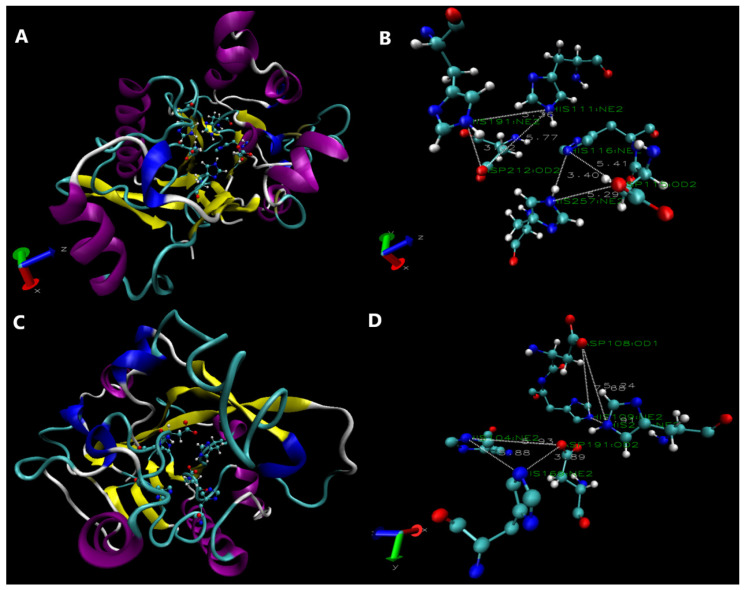
Three-dimensional structural simulation and functional prediction. (**A**) YtnP 3D structure; (**B**) magnified view of YtnP substrate-binding site; (**C**) AiiA 3D structure; (**D**) magnified view of AiiA substrate-binding site. α-helix (purple band), β-sheet (yellow arrow), random coil (blue band), and other structures (cyan and white tubes) are shown. Red, blue, cyan, and white represent Oxygen (red), nitrogen (blue), carbon (cyan), and hydrogen atoms (white), respectively. The residues and atoms of zinc-binding sites are labeled in the magnified view.

**Figure 6 microorganisms-13-01717-f006:**
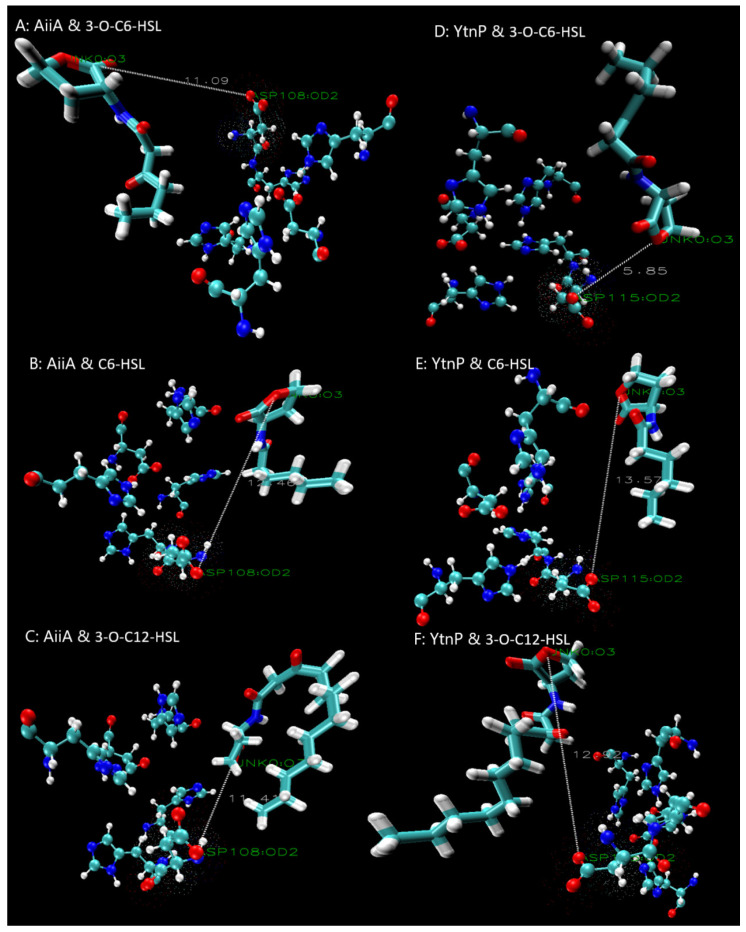
Ligand binding to AHL substrates. (**A**–**C**) Ligand pose of AiiA_DH82_ with 3-oxo-C_6_-HSL, C_6_-HSL and 3-oxo-C_12_-HSL; (**D**–**F**) ligand pose of YtnP_DH82_ with 3-oxo-C_6_-HSL, C_6_-HSLand 3-oxo-C_12_-HSL. Oxygen (red), nitrogen (blue), carbon (cyan), and hydrogen (white) atoms are shown. The distance between the active site (Dotted Asp residue) and the hydrolysis site of the ester oxygen is indicated by the white line.

**Figure 7 microorganisms-13-01717-f007:**
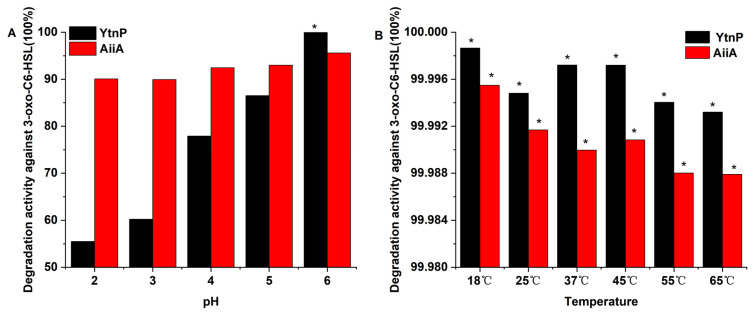
The enzyme’s activity in degrading AHL. The enzyme activity’s effect on AHL degradation was determined by the percentage of residual AHL(initial 300 µM in 1 mL reaction). Residual AHL was quantified via LC-MS (*m*/*z* 214). (**A**) indicates pH effects; (**B**) indicates temperature effects. Samples with residual AHL lower than 5 nM are marked with *.

**Table 1 microorganisms-13-01717-t001:** Sequences of PCR primers in this study.

Primer	Sequence
aiiA-F	5′-GGAATTCCATATGACAGTAAAGAAGCTTTATTTC-3′
aiiA-R	5′-CCGCTCGAGCGGTATATATATTCGAACACTTTACATCCCC-3′
ytnP-F	5′-GGAATTCATGGAGACATTGAATATTGGGAATTTTC-3′
ytnP-R	5′-CCGCTCGAGCGGTTTTTTCTCCCGTTGACAGATG-3′

Note: the sequences underline were respectively the restriction enzyme recognition sites of *Nde*I, *Xho*I and *Eco*RI.

**Table 2 microorganisms-13-01717-t002:** CDOCKER score of enzyme-substrate docking.

Substrate	AiiA_DH82_	YtnP_DH82_	*p* Value (*t*-Test)
3-OH-C_4_-HSL	21.898 ± 1.645	23.146 ± 1.705	0.113
C_4_-HSL	22.618 ± 1.848	23.050 ± 0.937	0.519
3-oxo-C_6_-HSL	27.820 ± 1.499	31.130 ± 0.804	1.389 × 10^−10^
C_6_-HSL	27.020 ± 1.561	28.396 ± 0.805	0.023
3-oxo-C_10_-HSL	36.196 ± 1.766	35.080 ± 1.702	0.049
3-oxo-C_12_-HSL	40.803 ± 1.175	38.229 ± 1.405	2.319 × 10^−7^

**Table 3 microorganisms-13-01717-t003:** Steady-state kinetic constants for the hydrolysis of 3-oxo-C_6_-HSL by lactonases.

	Km (mM)	kcat (s^−1^)	kcat/Km (M^−1^s^−1^)
AiiA_DH82_	7.8	1.616 × 10^3^	2.071 × 10^5^
YtnP_DH82_	15.33	3.840 × 10^3^	2.505 × 10^5^

Note: 100 µL of filter-sterilized 3-oxo-C_6_-HSL at concentrations of 200 µM, 250 µM, 300 µM, and 350 µM were mixed with 100 µL of 0.5 mg/mL purified enzyme at pH 6.5 and 28 °C for 45 min.

## Data Availability

The data presented in this study are available on request from the corresponding author due to privacy or ethical restrictions.

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
