# Peer review of "Development and Biochemical Characterization of Quorum Quenching Enzyme from Deep-Sea Bacillus velezensis DH82"

_microorganisms, 2025, doi:10.3390/microorganisms13081717_

Round 1

Reviewer 1 Report

Comments and Suggestions for Authors

Exploring the deep sea for possible lactonase enzymes which may have altered substrate ranges and stability for use in preventing QS by N-acyl-L-homoserine lactone systems is an interesting topic and the authors have developed an interesting approach. I found the bioinformatics probe, cloning, purification, and modeling portions of the paper pretty solid and relatively easy to follow. However, there are certain significant issues which need to be addressed before this paper should be considered again for evaluation. 

Significant concern

The work associated with and the conclusions drawn from Figure 7 are the most significant. The authors report the enzyme's efficiency remains stable (and at 100%) across a pH range from 7-11. Such stability would be rather unusual. More importantly it ignores that these molecules are highly prone to base-catalyzed hydrolysis and no such controls are presented. It is quite likely that at pH-values above 7 the lactone is hydrolyzed abiotically (no enzyme required). There are several works in this area over the last 15 years which could provide insight into this. Running an AHL only sample at these pH values would help distinguish this. It is likely the lactonases are active at some values in this range but at elevated pH natural hydrolysis is fast enough.

Moreover, it is not clear where the molecular weight of 245 comes from. The molecular weight of the ring closed AHL is 213. Hydrolysis would add approximately 18 for a total of 231, Typically the hydrolyzed 3-oxo-C6 is seen around 230-233 m/z. No reference is provided to account for this higher value. 

Finally, ions are added to test their effect on catalysis without justification or discussion. 

This figure feels very out of place from the rest of the paper and frankly could be removed entirely and the authors would still have a document worth consideration.

Minor concerns

-The language in the text can be difficult to follow and through editing is required to improve the readability of the document.

-The figure captions could use some increased details in some place to improve readability

-In figure 3 I think the lane descriptions are incorrect in the caption. 

Comments on the Quality of English Language

General concepts are easy to follow as are the methods and introduction. However, it would be a good idea for the authors to seek out an editor for clarity. 

Author Response

Response to Review Comments 

Dear Editor and Reviewers,

We would like to thank the reviewers for their kind comments and valuable suggestions. The manuscript has been carefully read through and the errors had been corrected. We submit here the revised manuscript with tracking marks, as well as a list of changes that refers to the comments. 

If you have any question about this paper, please don’t hesitate to let me know.

Sincerely yours,

Xiaohui Sun

Response to Reviewer 1's Comments

Reviewer 1’s comments:

Exploring the deep sea for possible lactonase enzymes which may have altered substrate ranges and stability for use in preventing QS by N-acyl-L-homoserine lactone systems is an interesting topic and the authors have developed an interesting approach. I found the bioinformatics probe, cloning, purification, and modeling portions of the paper pretty solid and relatively easy to follow. However, there are certain significant issues which need to be addressed before this paper should be considered again for evaluation. 

Significant concern

The work associated with and the conclusions drawn from Figure 7 are the most significant. The authors report the enzyme's efficiency remains stable (and at 100%) across a pH range from 7-11. Such stability would be rather unusual. More importantly it ignores that these molecules are highly prone to base-catalyzed hydrolysis and no such controls are presented. It is quite likely that at pH-values above 7 the lactone is hydrolyzed abiotically (no enzyme required). There are several works in this area over the last 15 years which could provide insight into this. Running an AHL only sample at these pH values would help distinguish this. It is likely the lactonases are active at some values in this range but at elevated pH natural hydrolysis is fast enough.

Answer: Thanks for the comments. We have consulted relevant literature as suggested, and also repeat the test of AHL residue at pH range above 7.0 , the results demonstrated that the AHL did alkaline hydrolyzed and non-detectable using LC-MS at the original m/z. Therefore, we modified this part in section 3.3 and replaced the figure 7B, also, related discussion was added with reference support in section 4.

Moreover, it is not clear where the molecular weight of 245 comes from. The molecular weight of the ring closed AHL is 213. Hydrolysis would add approximately 18 for a total of 231, Typically the hydrolyzed 3-oxo-C6 is seen around 230-233 m/z. No reference is provided to account for this higher value. 

Answer: Thanks for the comments. Yes, we did make a mistake about the mass-to-charge ratio to 3-oxo-C6. we used the molecular weight of the hydrogenated AHL at 214 for detection, and the value had been reversed in the manuscript.

Finally, ions are added to test their effect on catalysis without justification or discussion. 

This figure feels very out of place from the rest of the paper and frankly could be removed entirely and the authors would still have a document worth consideration.

Answer: Thanks for the comments. We removed the parts about the effects of ions as suggestion.

Minor concerns

-The language in the text can be difficult to follow and through editing is required to improve the readability of the document.

Answer: Thanks for the comments. The language errors had been revised in the revision.

-The figure captions could use some increased details in some place to improve readability

Answer: Thanks for the comments. The figure legends had been revised, including the wrong one in figure 3.

-In figure 3 I think the lane descriptions are incorrect in the caption. 

Answer: Thanks for the comments. Yes, the lane descriptions were incorrect, that had been revised in the revision.

Comments on the Quality of English Language

General concepts are easy to follow as are the methods and introduction. However, it would be a good idea for the authors to seek out an editor for clarity. 

Answer: Thanks for the comments. The manuscript had been carefully read through and the language had been improved.

Reviewer 2 Report

Comments and Suggestions for Authors

The manuscript reports the biochemical characterization of a N-acyl homoserine lactonase from Bacillus velezensis DH82 with quorum quenching activity. After the revision process, this manuscript could be considered for publication after a major revision. The following points were crucial for taking this decision:

General

  • Define abbreviations at their first mention (g. YtnP) to enhance readability and comprehension.
  • Merge [4], [5] (line 48) into [4,5], and [27], [28] into [27,28] (line 369).
  • Make sure that scientific names of species (g. Bacillus velezensis, P. aeruginosa and V. parahemolyticus, E. coli, lines 48, 51, 56) are written in italics, following standard formatting rules in scientific writing. Apply this rule throughout the entire manuscript, including the references.
  • Carefully check the manuscript for typographical and formatting errors to improve overall quality: P. aeruginosain (line 51),
  • Change the sentence to “Therefore, it was engineered for further study of its biological characteristics” (lines 208-209).
  • Unify the citation style throughout the manuscript. In the introduction section, references appear in both bracketed format and author–year format (lines 54-55). Choose only the citation style required by Microorganisms
  • Legends of figures should be presented as single paragraphs.
  • Please review the bibliography section carefully, as some references are missing the DOI, and in some cases, the month is provided instead of the issue number (line 509).

Results and discussion

  • In Figure 1, the letters appear slightly cut off in the multiple alignment. Please ensure that all labels and text elements are fully visible and clearly legible in the final version of the figure. The legend of Figure 1 should be presented as a single paragraph (lines 194-198).
  • Protein 1829 is described in the main text (line 190) but does not appear in the sequence comparison shown in Figure 1. Please include this protein in the figure or clarify its exclusion to ensure consistency between the text and the visual data.
  • The legend of Figure 2 should be presented as a single paragraph (lines 211-213) and placed on the same page as the figure.
  • The legend of Figure 3 appears to be incorrect, as lanes 1 and 2 should correspond to crude extracts, while lanes 3 and 4 likely represent the purified enzymes. Please verify and correct the figure legend to accurately reflect the content of each lane. The text refers to a sample loaded in lane 5 of the SDS-PAGE gel (page 8, line 220), but such a lane does not appear in the figure.
  • The statement about the zinc binding site in the legend of Figure 5 (lines 278-279) is not fully supported by the image, as the labels for the amino acid residues are not clearly visible. This figure 5, and also figure 6, should be improved to ensure that all labels are legible.
  • The symbol Kcat appears with an uppercase “K” (line 299) but it should be written with a lowercase “k” (kcat) as it represents a kinetic constant. Please correct this to align with standard biochemical nomenclature.
  • The sentence "The fewer residues of AHL refer to the stronger degradation activity of lactonases" (lines 316-317) is unclear and potentially misleading, as the term "residues" may be interpreted as amino acid residues. To avoid this confusion and to more accurately reflect the intended meaning, it is recommended to rephrase the sentence as: "A lower amount of remaining AHLs indicates stronger degradation activity by the lactonase”.
  • It is unclear whether the calibration curve shown in Figure 7A is truly necessary. Please consider whether it adds essential information to the results.
  • Section 3 is somewhat unclear, as the degradation of 3-oxo-C6-AHL appears to be nearly 100% in all cases. YtnPDH82 shows a higher 3-oxo-C6-AHL degradation rate than AiiADH82; however, it is not indicated whether this difference is statistically significant. Please include appropriate statistical analysis or clarify the significance of the observed differences to support the conclusions.
  • The ammonium ion is not correctly indicated in the text (line 336). To ensure chemical accuracy and clarity, it should be written as NH₄⁺, using the proper chemical notation.
  • Table 3 should not be split across two different pages. Please be sure that the entire table appears on a single page.
Comments on the Quality of English Language

Simplify complex sentences and ensure grammatical accuracy (lines 11-15).

Author Response

Response to Review Comments 

Dear Editor and Reviewers,

We would like to thank the reviewers for their kind comments and valuable suggestions. The manuscript has been carefully read through and the errors had been corrected. We submit here the revised manuscript with tracking marks, as well as a list of changes that refers to the comments. 

If you have any question about this paper, please don’t hesitate to let me know.

Sincerely yours,

Xiaohui Sun

Response to Reviewer 2's Comments

The manuscript reports the biochemical characterization of a N-acyl homoserine lactonase from Bacillus velezensis DH82 with quorum quenching activity. After the revision process, this manuscript could be considered for publication after a major revision. The following points were crucial for taking this decision:

General

Define abbreviations at their first mention (g. YtnP) to enhance readability and comprehension.

Answer: Thanks for the comments. The full name of Cluster of Orthologous Groups of proteins (COG) had been added. Also, the predicted lactonase, YtnP and AiiA, had been defined as YtnPDH82 and AiiADH82 with subscripts.

Merge [4], [5] (line 48) into [4,5], and [27], [28] into [27,28] (line 369).

Answer: Thanks for the comments. The number for reference citations had been merged to correct format.

Make sure that scientific names of species (g. Bacillus velezensis, P. aeruginosa and V. parahemolyticus, E. coli, lines 48, 51, 56) are written in italics, following standard formatting rules in scientific writing. Apply this rule throughout the entire manuscript, including the references.

Answer: Thanks for the comments. The font in the manuscript had been checked and revised, including the name of bacteria name in italics.

Carefully check the manuscript for typographical and formatting errors to improve overall quality: P. aeruginosain (line 51),

Answer: Thanks for the comments. The name of bacteria name had been revised in italics.

Change the sentence to “Therefore, it was engineered for further study of its biological characteristics” (lines 208-209).

Answer: Thanks for the comments. The sentence had been revised.

Unify the citation style throughout the manuscript. In the introduction section, references appear in both bracketed format and author–year format (lines 54-55). Choose only the citation style required by Microorganisms

Answer: Thanks for the comments. The citation format had been revised.

Legends of figures should be presented as single paragraphs.

Answer: Thanks for the comments. The figure legends had been revised to make the title and captions as single paragraphs.

Please review the bibliography section carefully, as some references are missing the DOI, and in some cases, the month is provided instead of the issue number (line 509).

Answer: Thanks for the comments. The format in bibliography section had been revised to be consistence.

Results and discussion

In Figure 1, the letters appear slightly cut off in the multiple alignment. Please ensure that all labels and text elements are fully visible and clearly legible in the final version of the figure. The legend of Figure 1 should be presented as a single paragraph (lines 194-198).

Answer: Thanks for the comments. The figure had been replaced with clear version, and the legend had been revised.

Protein 1829 is described in the main text (line 190) but does not appear in the sequence comparison shown in Figure 1. Please include this protein in the figure or clarify its exclusion to ensure consistency between the text and the visual data.

Answer: Thanks for the comments. The Protein 1829 and 3013 presented non-quorum quenching activity in our study, considering the consistence to the Phylogenetic tree analysis of the predicted enzymes in figure 2, we decided to delete these two predicted protein from the main text.

The legend of Figure 2 should be presented as a single paragraph (lines 211-213) and placed on the same page as the figure.

Answer: Thanks for the comments. The figure legends had been revised to make the title and captions as single paragraphs.

The legend of Figure 3 appears to be incorrect, as lanes 1 and 2 should correspond to crude extracts, while lanes 3 and 4 likely represent the purified enzymes. Please verify and correct the figure legend to accurately reflect the content of each lane. The text refers to a sample loaded in lane 5 of the SDS-PAGE gel (page 8, line 220), but such a lane does not appear in the figure.

Answer: Thanks for the comments. The figure legends and captions had been revised.

The statement about the zinc binding site in the legend of Figure 5 (lines 278-279) is not fully supported by the image, as the labels for the amino acid residues are not clearly visible. This figure 5, and also figure 6, should be improved to ensure that all labels are legible.

Answer: Thanks for the comments. The figure legends and captions had been revised. For the articulation in figure 5 and 6, the pixel seems to be compressed by Word software, we had replaced with a clear one, and submitted the original copy as attached pictures on the journal website.

The symbol Kcat appears with an uppercase “K” (line 299) but it should be written with a lowercase “k” (kcat) as it represents a kinetic constant. Please correct this to align with standard biochemical nomenclature.

Answer: Thanks for the comments. The symbol kcat had been revised with lowercase “k” in the revision.

The sentence "The fewer residues of AHL refer to the stronger degradation activity of lactonases" (lines 316-317) is unclear and potentially misleading, as the term "residues" may be interpreted as amino acid residues. To avoid this confusion and to more accurately reflect the intended meaning, it is recommended to rephrase the sentence as: "A lower amount of remaining AHLs indicates stronger degradation activity by the lactonase”.

Answer: Thanks for the comments. The sentence had been revised in the revision.

It is unclear whether the calibration curve shown in Figure 7A is truly necessary. Please consider whether it adds essential information to the results.

Answer: Thanks for the comments. The standard curve had been removed from the figure 7, and the figure captain had also been revised.

Section 3 is somewhat unclear, as the degradation of 3-oxo-C6-AHL appears to be nearly 100% in all cases. YtnPDH82 shows a higher 3-oxo-C6-AHL degradation rate than AiiADH82; however, it is not indicated whether this difference is statistically significant. Please include appropriate statistical analysis or clarify the significance of the observed differences to support the conclusions.

The ammonium ion is not correctly indicated in the text (line 336). To ensure chemical accuracy and clarity, it should be written as NH₄⁺, using the proper chemical notation.

Answer: Thanks for the comments. We have consulted relevant literature as suggested from other reviewer, and also repeat the test of AHL residue at pH range above 7.0 , the results demonstrated that the AHL did alkaline hydrolyzed and non-detectable using LC-MS at the original m/z at 214. Therefore, we modified this part in section 3.3 and replaced the figure 7B, also, related discussion was added with reference support in section 4. For the parts of irons affects, were removed as suggested by other reviewer.

Table 3 should not be split across two different pages. Please be sure that the entire table appears on a single page.

Answer: Thanks for the comments. The format of the table had been revised.

Reviewer 3 Report

Comments and Suggestions for Authors

Revision of the manuscript “Development and biochemical characterization of quorum 2 quenching enzyme from deep sea Bacillus velezensis DH82”.

The manuscript is from interest due to the searching for alternatives against aquatic pathogens and the abuse of antibiotics to control them, in this case the Quorum-quenching enzymes. Some observations are presented as follows:

Although COG notation has become a common term it must be defined.

In abstract, we understand that the YtnP is a QQ enzyme, but the author must be clarified before or define that the the homologous sequences with β-lactamase domain-containing protein were predicted as potential QQ enzyme, was named YtnP or explained.

Please wrote the names of bacteria in italics alomg the manuscript.

In section 2.4 the components of LB medium must be write in lower case (Tryptone, Yeast, etc.). Also, in this section the numbers of the molecule NaH2PO4 must be in subscript notation.

Line 113 and 124, 145, please write the quantity as a word instead numerical form.

In section 3.2 second paragraph, the author described the SDS-Page and mentioned… the His-tagged AiiADH82 at 31.99 kDa and YtnPDH82 at 35.82 kDa were observed in lane 3 and lane 5 respectively…Although, the gel in Fig. 3 has 5 lines, the authors must describe de corresponding line (M, 1, 2,3,4). Please, point with an arrow for a better explanation because is difficult to appreciate the 31.99 kDa (AiiADH82) and 35.82 kDa (YtpnPDH82), is confusing.

Author Response

Response to Reviewer 3's Comments

The manuscript is from interest due to the searching for alternatives against aquatic pathogens and the abuse of antibiotics to control them, in this case the Quorum-quenching enzymes. Some observations are presented as follows:

Although COG notation has become a common term it must be defined.

In abstract, we understand that the YtnP is a QQ enzyme, but the author must be clarified before or define that the the homologous sequences with β-lactamase domain-containing protein were predicted as potential QQ enzyme, was named YtnP or explained.

Answer: Thanks for the comments. The full name of Cluster of Orthologous Groups of proteins (COG) had been added in the abstract. And the predicted YtnP and AiiA had been defined as YtnPDH82 and AiiADH82 with subscripts.

Please wrote the names of bacteria in italics alomg the manuscript.

Answer: Thanks for the comments. The font in the manuscript had been checked and revised, including the name of bacteria name.

In section 2.4 the components of LB medium must be write in lower case (Tryptone, Yeast, etc.). Also, in this section the numbers of the molecule NaH2PO4 must be in subscript notation.

Answer: Thanks for the comments. The mistakes had been revised in the manuscript.

Line 113 and 124, 145, please write the quantity as a word instead numerical form.

Answer: Thanks for the comments. The quantity of the solutions had been revised with final concentration as word form as suggestion.

In section 3.2 second paragraph, the author described the SDS-Page and mentioned… the His-tagged AiiADH82 at 31.99 kDa and YtnPDH82 at 35.82 kDa were observed in lane 3 and lane 5 respectively…Although, the gel in Fig. 3 has 5 lines, the authors must describe de corresponding line (M, 1, 2,3,4). Please, point with an arrow for a better explanation because is difficult to appreciate the 31.99 kDa (AiiADH82) and 35.82 kDa (YtpnPDH82), is confusing.

Answer: Thanks for the comments. Yes, the lane descriptions were incorrect, that had been revised in the revision.

Round 2

Reviewer 2 Report

Comments and Suggestions for Authors

After the revision process, this manuscript could be considered for publication after a minor revision.

Please revise again the bibliography section carefully, as there are still some references whose DOI are missing. We have noticed that the requested formatting changes regarding the merging of bracketed references (when there are more than two) have not been addressed.

Author Response

Please revise again the bibliography section carefully, as there are still some references whose DOI are missing. We have noticed that the requested formatting changes regarding the merging of bracketed references (when there are more than two) have not been addressed.

Answer: Thanks for pointing out the mistakes. The reference citation had been checked and the billiography section had also been revised carefully, including the DOI, font formation of authors' name and Bacterial name.